# Ultrasound-controllable engineered bacteria for cancer immunotherapy

Mohamad H. Abedi [1,5,6], Michael S. Yao [1,6], David R. Mittelstein[2], Avinoam Bar-Zion[3], Margaret B. Swift[3], Audrey Lee-Gosselin[3], Pierina Barturen-Larrea [3], Marjorie T. Buss [3] & Mikhail G. Shapiro [3,4 ✉]

Rapid advances in synthetic biology are driving the development of genetically engineered microbes as therapeutic agents for a multitude of human diseases, including cancer. The immunosuppressive microenvironment of solid tumors, in particular, creates a favorable niche for systemically administered bacteria to engraft and release therapeutic payloads. However, such payloads can be harmful if released outside the tumor in healthy tissues where the bacteria also engraft in smaller numbers. To address this limitation, we engineer therapeutic bacteria to be controlled by focused ultrasound, a form of energy that can be applied noninvasively to specific anatomical sites such as solid tumors. This control is provided by a temperature-actuated genetic state switch that produces lasting therapeutic output in response to briefly applied focused ultrasound hyperthermia. Using a combination of rational design and high-throughput screening we optimize the switching circuits of engineered cells and connect their activity to the release of immune checkpoint inhibitors. In a clinically relevant cancer model, ultrasound-activated therapeutic microbes successfully turn on in situ and induce a marked suppression of tumor growth. This technology provides a critical tool for the spatiotemporal targeting of potent bacterial therapeutics in a variety of biological and clinical scenarios.

[1] Division of Biology and Biological Engineering, California Institute of Technology, Pasadena, CA 91125, USA. [2] Division of Engineering and Applied Sciences, California Institute of Technology, Pasadena, CA 91125, USA. [3] Division of Chemistry and Chemical Engineering, California Institute of Technology, Pasadena, CA 91125, USA. [4] Howard Hughes Medical Institute, California Institute of Technology, Pasadena, CA 91125, USA. [5]Present address: Department of Biochemistry, Institute for Protein Design and Howard Hughes Medical Institute, University of Washington, Seattle, WA 98195, USA. [6]These authors contributed equally: Mohamad H. Abedi, Michael S. Yao. ✉email: mikhail@caltech.edu

Cell therapies are rapidly emerging as an exciting and effective class of technologies for cancer treatment[1–3]. Among the cell types being investigated for therapy, immune cells have excelled in the treatment of hematologic malignancies. However, their use in solid tumors has been hampered by their reduced ability to penetrate and function in the tumor's immunosuppressive environment, especially within immune-privileged hypoxic cores[4–6]. Conversely, the reduced immune activity of some tumor cores creates a favorable microenvironment for the growth of certain bacteria, which can reach the tumors after systemic administration[7–9]. Capitalizing on their tumor-infiltrating properties, such bacteria can be engineered to function as effective cellular therapies by secreting therapeutic payloads to directly kill tumor cells or remodel the microenvironment to stimulate anti-tumor immunity[10–15]. However, the benefits of microbial therapy are often counterbalanced by safety concerns accompanying the systemic injection of microbes into patients with limited control over their biodistribution or activity[1,16,17]. This is especially important given the well-documented engraftment of circulating bacteria into healthy tissues such as the liver, spleen, and certain hypoxic stem cell niches[18–21]. To avoid damaging healthy organs, it is crucial that the therapeutic activity of microbes be targeted to tumors.

Among the available mechanisms to regulate microbial function, systemically administered chemical inducers[20,22–26] are convenient to apply but incapable of targeting a particular anatomical site. Meanwhile, light-induced control elements provide high spatiotemporal precision[27–29], but are constrained by the poor penetration of light into intact tissues[30]. Radiation-induced promoters can be targeted by deeply penetrant energy. However, ionizing radiation carries the risk of damage to host immune cells and engineered microbial cells[31]. Alternatively, temperature-based transcriptional regulators enable spatiotemporal control at depth, since temperature can be elevated precisely within a well-tolerated range[32,33] in deep tissues using noninvasive methods such as focused ultrasound (FUS)[34–36].

Indeed, it was recently demonstrated that FUS can be used in conjunction with temperature-dependent repressors to control the expression of bacterial genes[37]. However, these repressors operated in clinically irrelevant cloning strains of bacteria, had non-therapeutic outputs, and produced only transient activation unsuitable for tumor treatment, which typically requires weeks of therapeutic activity[10,11,22].

Here we describe the development of FUS-activated therapeutic bacteria in which a brief thermal stimulus activates sustained release of anti-cancer immunotherapy. We engineer these cellular agents by adapting temperature-sensitive repressors to the tumor-homing probiotic species *E. coli* Nissle 1917[10,38] and designing gene circuits in which they control an integrase-based state switch[37,39] resulting in long-term therapy production. To improve the safety and efficacy of these cells, we screen random and rationally designed libraries of gene circuit variants for constructs with minimal baseline activity and maximal induction upon thermal stimulation. We use the optimized gene circuits to express immune checkpoint inhibitors targeting CTLA-4 and PD-L1. In a mouse cancer model, we show that the resulting engineered microbes are reliably and chronically activated by a brief, noninvasive FUS treatment after systemic administration to release therapy and successfully suppress tumor growth.

## Results

### Characterizing thermal bioswitches in a therapeutically relevant microbe.
To develop a temperature-actuated therapeutic circuit, we started with high-performance temperature-dependent transcriptional repressors, which actuate transient gene expression in response to small changes in temperature around 37 °C[37]. Since genetic elements tend to behave differently across cell types due to variations in protein expression and other aspects of the intracellular environment[40], we first characterized the performance of these repressors in our chosen therapeutic chassis: *E. coli* Nissle 1917 (EcN). This bacterial strain is approved for human probiotic use and is commonly employed in microbial tumor therapy[10,38]. We selected three repressor candidates—TlpA39, TcI, and TcI42—as our starting points due to their desirable activation temperature thresholds of 39 °C, 38 °C, and 42 °C, respectively. In its natural host, *Salmonella typhimurium*, TlpA is speculated to be responsible for the regulation of virulence genes upon entry into a warm host organism[41]. Meanwhile, TcI is a temperature-sensitive mutant of the bacteriophage lambda protein "cI"[42]. In its native context, cI serves as a transcriptional repressor that allows the bacteriophage lambda virus to establish and maintain latency[43].

To evaluate the performance of these candidates we designed reporter constructs where they regulate the expression of a green fluorescent protein (GFP) (Fig. 1a), transformed them into EcN cells, and measured the corresponding cell density-normalized fluorescence intensity as a function of temperature between 33 °C and 42 °C (Fig. 1b). This construct provided a tractable platform for us to directly evaluate the inducibility of thermally responsive repressors in EcN cells by measuring GFP fluorescence. As we were interested in using these bioswitches in vivo, we focused on the fold-change between the mammalian physiological temperature (37 °C) and an elevated temperature that can be used to trigger activation in vivo while minimizing thermal damage to local tissues (42 °C) (Fig. 1c). Results from these experiments indicated that TcI42 is the best candidate for integration into our thermal switch since it exhibits strong induction at 42 °C while maintaining low levels of baseline activity.

With TcI42 serving as the thermal transducer in our cells, we next sought to determine the minimal heating duration and ideal heating parameters required to achieve strong activation while minimizing damage to cells. We stimulated cells carrying the circuit described in Fig. 1a by elevating the temperature to 42 °C for different durations and measured the corresponding fluorescence intensity (Fig. 1d). The results indicated that a minimal heating time of one hour is needed for robust activation. We quantified the effect of this thermal dose on microbial cell viability and simultaneously tested a pulsatile heating scheme that was previously shown to enhance viability in mammalian cells[44]. For the pulsatile heating scheme, the duty cycle was kept constant at 50% while alternating the temperature between 37 °C and 42 °C, resulting in a total of one hour at 42 °C over a two-hour period, with pulse duration varying between 1 and 60 min (Fig. 1e). As hypothesized, cell viability decreased as the pulse duration increased, while induction levels did not significantly vary (Fig. 1f). Based on these results, we selected a five-minute pulse duration for subsequent applications, as this heating paradigm enhanced cell viability while being readily achievable with a focused ultrasound setup. Collectively, our experiments identified and characterized TcI42 as an effective thermal transducer to control gene expression in the therapeutically relevant EcN strain.

### Constructing a thermally actuated state switch.
On its own, the TcI42 switch is not sufficient for microbial cancer therapy. This switch is transiently activated for the duration of heating, while tumor therapy requires weeks to effectively suppress tumor growth. Since daily FUS application over this period may be undesirable in a clinical setting, we set out to engineer a gene circuit that maintains a prolonged therapeutic response following a single, brief thermal activation.

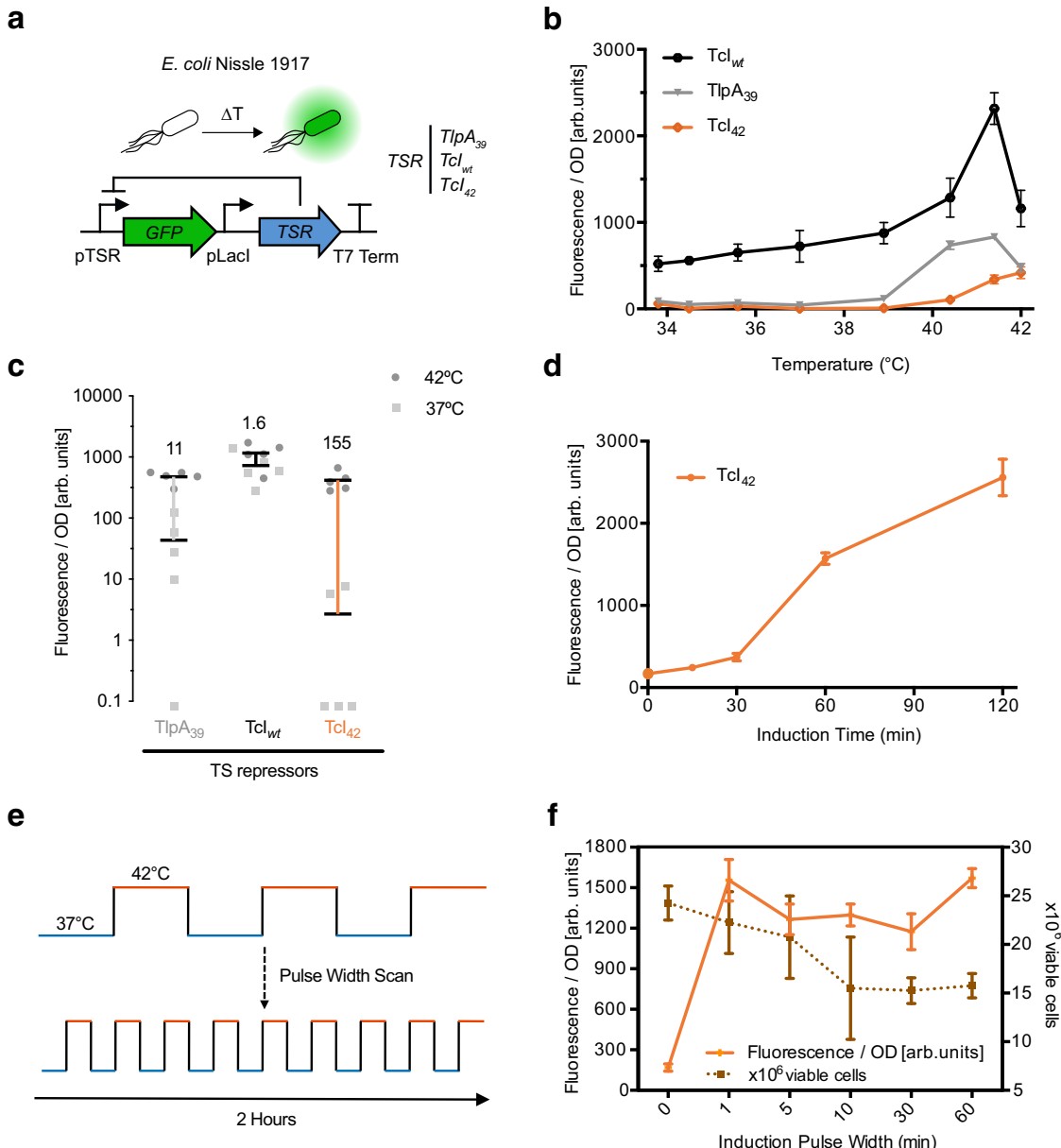

**Fig. 1 Evaluating temperature-sensitive transcriptional repressors in *E. coli* Nissle 1917. a** Illustration of the genetic circuit used to characterize the behavior of temperature-sensitive repressors in *E. coli* Nissle 1917. **b** Optical density (OD$_{600}$)-normalized fluorescence as a function of induction temperature for a fixed duration of 1 h, measured 24 h after induction. Error bars represent ±SEM. To confirm that the resulting data is not driven by temperature driven changes to OD, wildtype EcN were similarly analyzed and displayed no temperature dependent fold change. Additionally, total cell count by flow cytometry was also used as a proxy for cell number and generated similar results to the ones collected by normalizing through OD as a proxy for cell count (Supplementary Fig. 1). **c** OD-normalized fluorescence 24 h after a 1-hour induction at 37 °C or 42 °C for the constructs shown in (**b**). Measurements with values below the bottom of the y-axis appear below the axis. Bars indicate the mean. Vertical lines indicate the difference between the 42 °C and 37 °C conditions. Numbers indicate fold-change. **d** OD-normalized fluorescence as a function of induction duration. Cells were stimulated at 42 °C and fluorescence measured 24 h later. **e** Illustration of the pulsatile heating scheme used to optimize thermal induction and cell viability. **f** OD-normalized fluorescence as a function of pulse duration for the TcI$_{42}$ circuit. All samples were stimulated for a total of 1 h at 42 °C and 1 h at 37 °C and evaluated 24 h later. Viable cell counts at various pulse durations plotted to reflect cell viability. Where not seen, error bars (±SEM) are smaller than the symbol. $n = [5, 5, 6, 4]$ biologically independent replicates for panels [**b**, **c**, **d**, **f**]. All source data are provided as a Source Data file.

To enable stable thermal switching, we placed the expression of Bxb1, a serine integrase, under the control of the pL/pR phage lambda thermally inducible promoters, whose activity is regulated by the TcI42 repressor (Fig. 2a). Serine integrases such as Bxb1 were initially discovered in bacteriophages as a class of enzymes targeting DNA sequences known as attP and attB sites. In their native context, these sites serve as a hub for the integration of bacteriophage DNA into the genome of target cells[45]. However,

these sites can be repurposed to flank an arbitrary DNA sequence and mediate its inversion, resulting in a stable switching functionality[46]. Our design combines the temperature sensitivity of TcI42 with this permanent effector function of the Bxb1 integrase. At physiological temperatures of approximately 37 °C, constitutive expression of the TcI42 repressor from the pLacI promoter represses the expression of Bxb1. Upon thermal stimulation, the release of TcI42 repression results in a burst of

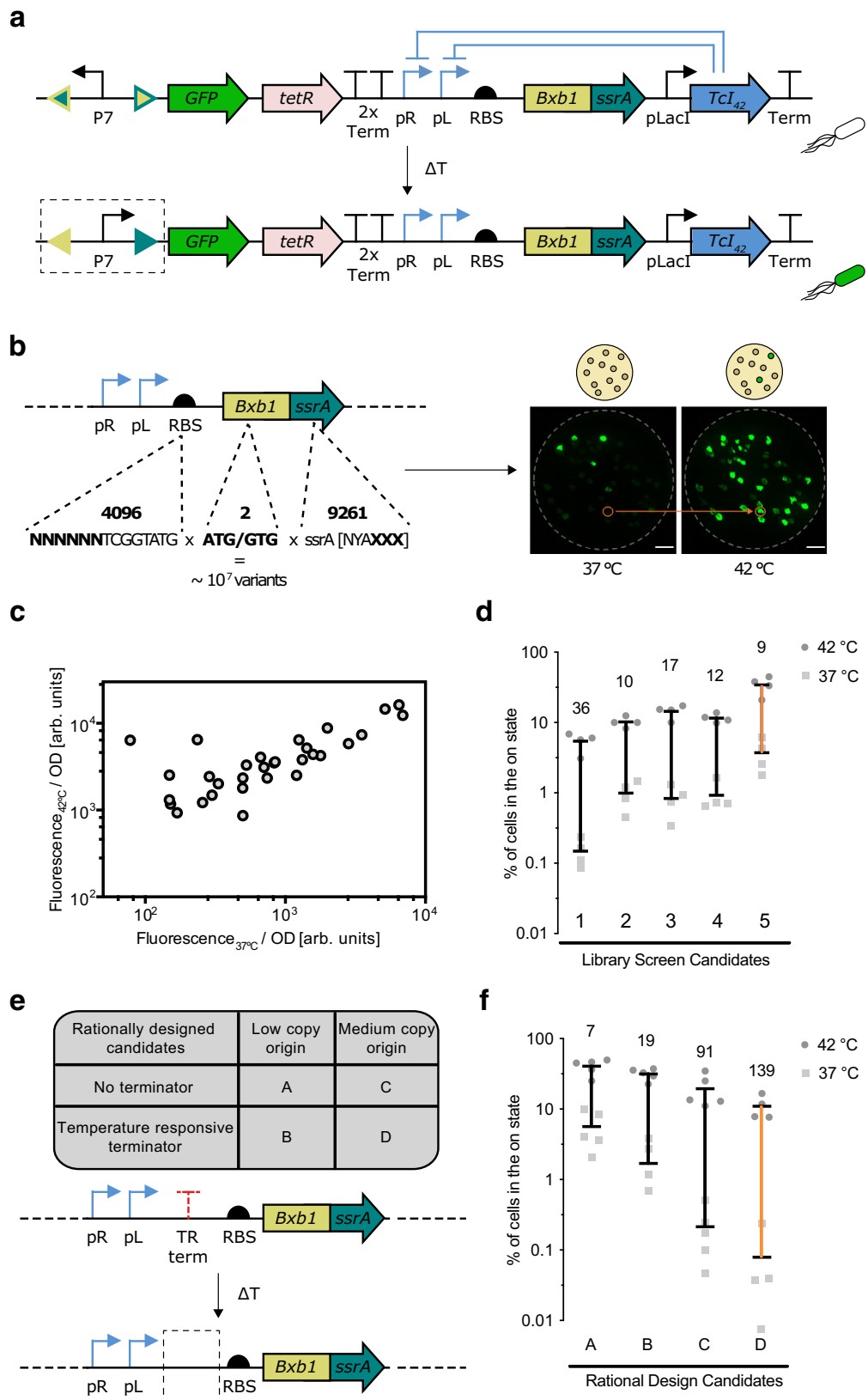

Bxb1 expression. Thermally derepressed Bxb1 expression then catalyzes the inversion of minimal recognition sites attP and attB flanking the P7 promoter, resulting in subsequent expression of a fluorescent reporter to monitor the state of the circuit and a tetracycline resistance cassette serving as a placeholder for a

therapeutic protein (Fig. 2a). Because the attP and attB recognition sites are modified post-inversion by the Bxb1 enzyme, the inverted DNA sequence is not recognized by subsequent Bxb1 interaction and is therefore a permanent inversion event. Subsequently, the P7 promoter continues to drive the expression

**Fig. 2 Construction and optimization of a temperature responsive state switch. a** Illustration of the genetic circuit constructed to establish a temperature responsive state switch. TetR is the tetracycline resistance cassette. **b** Illustration of the sites targeted in a high throughput screen to optimize circuit switching. A representative fluorescence image of replica plates used to screen for circuit variants. Plates were incubated at the indicated temperature for one hour and further incubated at 37 °C until colonies grew large enough for analysis. The orange circle indicates an example colony selected for further assay. **c** Circuit variants from the screen in (**b**) characterized for their fluorescence at 37 °C and 42 °C. **d** Percent conversion to the on-state 24 h after a 1-hour thermal stimulation at 42 °C or 37 °C for five of the circuit variants from (**c**). Bars indicate the mean. Vertical lines indicate the difference between the 42 °C and 37 °C conditions. Numbers indicate fold-change. **e** Summary of rational modifications made to reduce leakage in the circuit at 37 °C. **f** Percent induction 24 h after a 1-hour of thermal induction at 42 °C compared to baseline incubation at 37 °C for four circuit variants described in **e**. Measurements with values below the bottom of the y-axis appear below the axis. Bars indicate the mean. Vertical lines indicate the difference between the 42 °C and 37 °C conditions. Numbers indicate fold-change. $n = [4, 5]$ biologically independent replicates for panel [**d**, **f**]. All source data are provided as a Source Data file.

of its protein payloads even when the temperature stimulus is terminated. The P7 promoter was chosen from a depository of synthetic constitutive bacterial promoters[47] due to its balance of strongly driving the expression of a genetic payload without creating excessive stress to the cell. To avoid unregulated expression of Bxb1 we insulated the activity of the temperature-activated promoter by inserting two strong terminators upstream to block activity from other regions of the plasmid[48].

The ideal performance of the circuit described above would maintain low baseline activity at physiological temperature while providing strong and lasting induction once thermally stimulated. To achieve this performance, we tuned three key sequence elements affecting Bxb1 translation and stability: the Bxb1 ribosomal binding sequence (RBS), start codon, and ssrA degradation tag (Fig. 2b). The ssrA tag is a short peptide that naturally gets added to the C terminus of proteins whose translation has stalled. Proteins that carry this sequence as a fusion are targeted for degradation by endogenous bacterial proteases[49]. To efficiently identify the optimal versions of these elements we performed a library screen that consisted of randomized 6-bp sequences within the Bxb1 RBS, two Bxb1 start codon choices, and randomized terminal tripeptides in the Bxb1 ssrA degradation tag[50]. Two start codons were tested because the non-canonical start codon GUG can down-regulate ribosomal efficiency, and the last three amino acids of the ssrA degradation tag were randomized because they strongly modulate the degradation rate of ssrA-tagged proteins[49]. A total landscape of approximately $10^7$ possible unique variants was sampled using a high-throughput plate-replication assay (Fig. 2b). Agar plates containing colonies of library members were first replicated, and then one plate was incubated at 37 °C to assess baseline expression, while the other plate was stimulated at 42 °C for an hour and then returned to 37 °C for the rest of the growth period. The temperature-dependent fluorescence of a representative sampling of variants is shown in Fig. 2c. We selected a subset of variants with low leak and high activation to quantify their switching performance with a larger number of replicates (Fig. 2d). Out of these candidates, we selected candidate #5 for further optimization since it activated the largest percentage of the cells upon stimulation, a metric that is important to ensure strong therapeutic activity in vivo, while still retaining a reasonable temperature-dependent fold change (Fig. 2d).

To reduce the baseline activity of candidate #5, we modified two additional circuit components (Fig. 2e). The first modification changed the origin of replication from the low-copy origin pSC101 to the medium-copy origin p15A. The second modification explored the effect of inserting a temperature-sensitive terminator upstream of the Bxb1 coding sequence. This family of terminators have been engineered to mimic temperature-modulated structures known as RNA thermometers that are found in the 5' untranslated region of microbial mRNAs and play an important role in regulating microbial gene expression in

response to temperature changes[51]. In our circuit, we used this terminator to introduce a temperature-sensitive secondary structure in the mRNA transcript that helps terminate protein expression at low temperatures, adding to the control provided by TcI42 to prevent leaky Bxb1 protein production at physiological temperature[52]. At 42 °C, this terminator loses its secondary structure and Bxb1 expression is unimpeded. We assessed the performance of four constructs with either one or both of these modifications (Fig. 2f).

Increasing the copy number of the plasmid and inserting the terminator reduced baseline activation independently. When combined together, these modifications resulted in significantly reduced leakage while maintaining a large fold-change in activated cells upon induction. The resulting construct, obtained through a combination of randomized and rational engineering, displayed a more than 100-fold change in activity between 37 °C and 42 °C.

**Engineering cells for thermally actuated secretion of immunotherapy.** To demonstrate the functionality of our optimized thermally-actuated cells in a clinically relevant scenario, we modified the output of their gene circuit to express an anti-tumor therapeutic payload (Fig. 3a). We selected αCTLA-4 and αPD-L1 nanobodies, which block signaling through the CTLA-4 and PD-L1 checkpoint receptor pathways, that are heavily implicated in T-cell silencing within immunosuppressive solid tumors. Checkpoint inhibitors such as αCTLA-4 and αPD-L1 have emerged as a major class of cancer therapy, but their therapeutic efficacy is commonly accompanied by the risk of unintentionally activating autoimmunity in bystander tissues when administered systemically[53,54]. By combining the ability of FUS to target specific areas deep within tissues with the tumor infiltration, thermal response and molecular specificity of our engineered cells, we reasoned that we could target the activity of these potent immunomodulators to tumors and thereby mitigate the risks of systemic exposure.

αCTLA-4 and αPD-L1 have been shown to produce antitumor effects when released by tumor-injected probiotics[10]. We hypothesized that local FUS-activated release of these proteins in tumors from systemically administered engineered bacteria would suppress tumor growth. To test this hypothesis, we fused αCTLA-4 and αPD-L1 to a PelB secretion tag to enhance their extracellular release upon activation and cloned each construct in place of the tetracycline cassette in our optimized switching circuit. The pelB leader peptide, derived from the *Erwinia carotovora* pelB gene, has been previously used to secrete proteins from microbes[22,55,56]. In addition, to stabilize our plasmids for long-term retention in vivo without antibiotic selection, we added an Axe-Txe toxin-antitoxin stability domain, which ensures retention of the plasmid in a cell population by eliminating cells that lose it[57,58]. The Axe-Txe type II toxin anti-toxin system

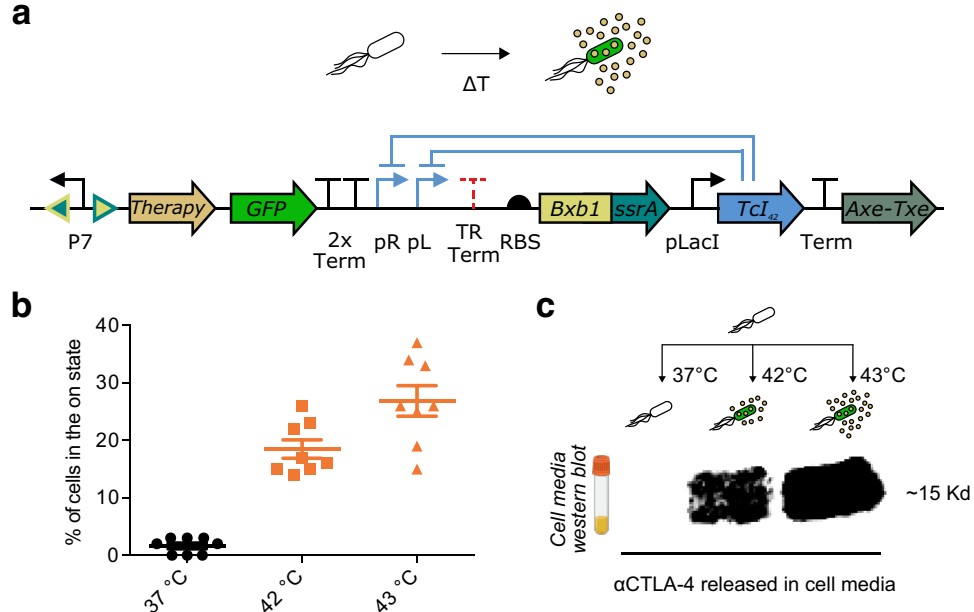

**Fig. 3 Thermally activated sustained release of a therapeutic payload. a** Temperature responsive state switch modified to release αCTLA-4 or αPD-L1 nanobodies. The circuit includes an Axe-Txe stability cassette. **b** Percent activation 24 h after a 1-hour of thermal induction at 37 °C, 42 °C or 43 °C for the circuit described in (**a**). Error bars represent (±SEM). **c** Western blot against hexahistidine-tagged αCTLA-4 nanobodies. Cells were induced for 1 h at 37 °C, 42 °C or 43 °C, then expanded in 5 ml of media for 24 h at 37 °C before collecting the media and assaying for the release of αCTLA-4 nanobodies. The original western blot image is shown in Supplementary Fig. 2. Similar staining was done to confirm αPD-L1 release. $n = 8$ biologically independent replicates for (**b**). All source data are provided as a Source Data file.

originates from the Axe-Txe locus of the gram-positive *Enterococcus faecium* plasmid pRUM[57].

The thermal switching functionality of our therapeutic circuits closely resembled their non-therapeutic counterpart. The circuit containing αCTLA-4 maintained a tight off-state at 37 °C while exhibiting robust fold-changes upon induction at 42 °C and 43 °C (Fig. 3b). Furthermore, upon tracking induced cells post-induction we saw no evidence of mutational escape, suggesting a tolerable level of burden[59] (Supplementary Fig. 3).

To assess the secretion of therapeutic nanobodies upon activation, we stimulated the cells for one hour at 37 °C, 42 °C and 43 °C, then cultured them for one day at 37 °C and performed a Western Blot to evaluate the levels of αCTLA-4 nanobodies released in their media. This experiment demonstrated that αCTLA-4 nanobodies are reliably secreted exclusively upon stimulation at 42 °C and 43 °C (Fig. 3c). We could not detect any secretion when the cells were incubated at 37 °C. Similar characterization was performed for cells expressing αPD-L1.

**Focused ultrasound activation elicits in vivo tumor suppression.** To enable thermal control of engineered therapeutic microbes in vivo we built a FUS stimulation setup providing feedback-controlled pulsatile tumor heating (Fig. 4a), capturing the key features of clinically available instruments[60,61]. We demonstrated that our system is capable of toggling the temperature in the tumor of a live animal between 37 °C and 43 °C every five minutes (Fig. 4a, Supplementary Fig. 4). We set the focal maximum temperature inside the tumor at 43 °C to allow more of the mass to be heated above 42 °C and ensure reliable activation within the context of a mouse. While this could lead to some thermal damage, we reasoned that such damage within the tumor is acceptable and could synergize with the microbial immunotherapy[62,63].

Using this in vivo setup, we tested our ability to locally activate systemically administered therapeutic microbes inside tumors. We seeded $5 \times 10^6$ A20 murine tumor cells in the right flanks of

BALB/c mice (Fig. 4b). Once the tumors grew to approximately 100 mm³, we intravenously injected $10^8$ EcN cells comprising a 1:1 mixture of cells engineered for thermally actuated αCTLA-4 or αPD-L1 secretion. This combination therapy was chosen because it provides a stronger anti-tumor effect compared to either therapeutic output on its own[10] (Supplementary Fig. 5). Injected microbes were given two days to engraft in tumors before they were stimulated with FUS. After FUS activation, tumor growth was monitored to assess therapeutic efficacy.

We observed major retardation in tumor growth in FUS-treated tumors colonized by therapeutic cells, while growth rates in controls including non-FUS treated mice, animals treated with only FUS, and subjects injected with wild-type EcN were substantially higher (Fig. 4c, Supplementary Fig. 6). The observed effect on tumor growth was comparable to that obtained by systemically administering antibody-based immune checkpoint inhibitors against αCTLA-4 and αPD-L1, or by systemically injecting pre-activated therapeutic EcN. However, unlike each of these established treatments, whose activity depends solely on the systemic biodistribution of the injected agents and thus carries potential for side-effects, FUS-activated bacterial therapy can act in a spatially localized fashion. To illustrate this localization, after completing this experiment, we collected tumors, livers, and spleens, chemically homogenized them, and plated the suspension on selective media. By counting the percentage of activated bacteria, we first demonstrated that our thermal switch is triggered in targeted tumors and remains active for at least two weeks post-activation (Fig. 4d). We then examined the percentage of activated bacterial agents in the tumors, livers and spleens of FUS-treated animals, which confirmed that our activation is primarily localized to targeted tumors while sparing bystander tissues (Fig. 4e).

One of the six FUS-activated tumors disappeared as a result of the treatment, and bacterial activation inside it could not be quantified. This tumor does not represent the typical outcome of this therapy. In three out of nine FUS-treated tumors, ultrasound

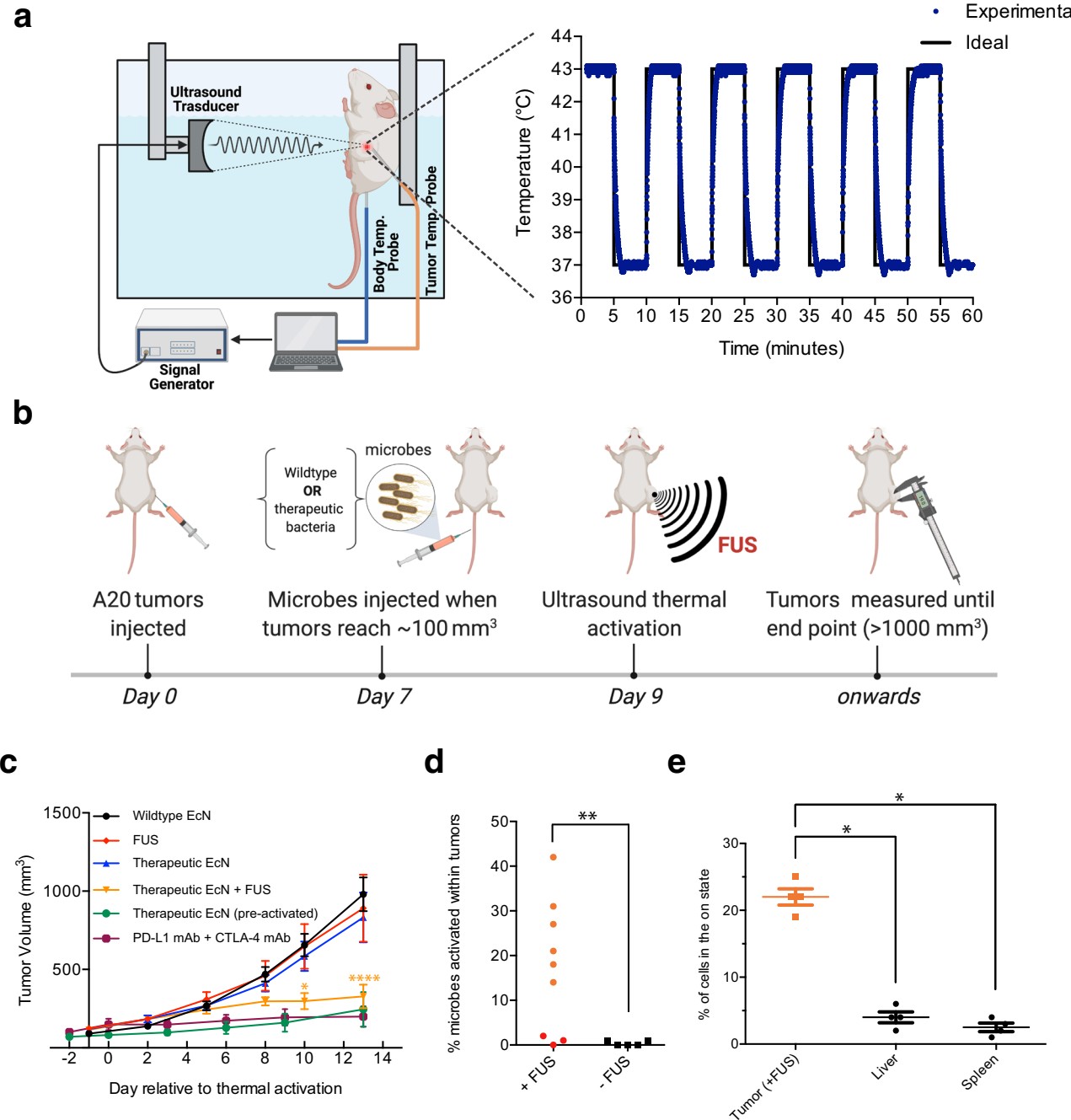

failed to activate the therapeutic bacterial circuit. This could be due to limitations in our heating setup, which is currently capable of only partially heating the tumor mass. Since EcN have been shown to colonize discrete regions within A20 tumors[64], these microbes could have been poorly heated if their locations did not coincide with the ultrasound focus (Supplementary Fig. 7). Clinical FUS systems that use MRI feedback and dynamic mechanical or electrical focusing to ensure the precise heating of defined volumes can overcome these limitations by heating the whole tumor[65]. The three non-activated mice were removed from our analysis of tumor growth. However, even without excluding these animals, the therapeutic EcN and FUS-treated mice show a significant reduction in endpoint tumor volume compared to mice treated with only therapeutic EcN or only FUS, with p-values of 0.0002 and 0.0001, respectively in a Dunnett's multiple comparisons test. Overall, our in vivo experiments demonstrated

that EcN cells engineered for thermally controlled checkpoint inhibition are able to home to and engraft in tumors from systemic circulation, become activated specifically in response to FUS, maintain this activity for at least two weeks after a 1-hour FUS treatment and significantly reduce tumor growth.

## Discussion

Our results establish a system for targeted probiotic immunotherapy that couples the special ability of therapeutic bacteria to home into the necrotic core of solid tumors with the capacity of FUS to locally activate their therapeutic function. The sustained activation of these therapeutic bacteria is enabled by a thermal state switch developed through high throughput genetic engineering to have low baseline activity, rapid induction upon stimulation and sustained activity in situ. When this state switch

**Fig. 4 Ultrasound-activated bacterial immunotherapy reduces tumor growth in vivo. a** Illustration of the automated setup used to deliver FUS hyperthermia to tumors (left) and representative time course of tumor temperature from a mouse treated with alternating 5-min steps between 37 °C and 43 °C. **b** Diagram illustrating the experiment performed to assess the activation of microbial antitumor immunotherapy in vivo. Mice were injected with a 1:1 mixture of EcN cells carrying the αCTLA-4 or αPD-L1 circuits, or wildtype EcN. EcN cells were washed and adjusted to 0.625 $OD_{600}$ before injecting 100 μL per mouse intravenously. Ultrasound was applied for a total of 1 h at 43 °C with 50% duty cycle and 5-min pulse duration. **c** Tumor sizes measured over two weeks in mice treated with wildtype EcN, therapeutic microbes in the absence of FUS, therapeutic microbes and FUS treatment, or FUS treatment alone. Asterisk represents statistical significance calculated with two-way ANOVA analysis where the therapy was compared to each of the controls with a Dunnett's multiple comparisons test (one-sided). * plotted, $p = [0.004(**), 0.0384(*), 0.0083(**)]$ when compared to [Wildtype, Therapeutic, FUS]; **** plotted, $p = [<0.0001$ for all]. **d** Percent activation of therapeutic EcN isolated from FUS-treated (nine mice) and non-FUS-treated (five mice) tumors two weeks after FUS treatment. ** plotted represents a $p$ value of 0.0085 when measured with a one-tailed Mann Whitney test. One of the FUS-activated tumors disappeared after treatment and bacterial activation inside it could not be quantified. Results in (**c, d**) were collected from four independent experiments conducted on separate days with new cells transformed for each. Where not seen, error bars (±SEM) are smaller than the symbol. Eight mice were analyzed for the [therapeutic EcN, FUS] groups, ten mice for the wildtype group, four for the therapeutic EcN (pre-activated), and six mice for the PD-L1 mAb + CTLA-4 mAb group. Ten mice were analyzed for the therapeutic condition, where three failed to activate. Data from nine therapeutic mice is displayed in panel (**d**) since in the tenth the tumor disappeared and couldn't be analyzed. **e** Background activation in bystander tissues following FUS activation of tumors. Percent activation of therapeutic EcN isolated from FUS-treated tumors and bystander organs (liver, spleen). *$p = 0.0143$ (tumor vs liver) and 0.0143 (tumor vs spleen). Statistical analysis was done with a Mann Whitney test (one-sided). Four mice were analyzed in panel (**e**). All source data are provided as a Source Data file.

is used to actuate the release of immune checkpoint inhibitors, the resulting engineered microbes can be activated inside tumors by brief FUS exposure to secrete their therapeutic payload over an extended timeframe and substantially reduce tumor growth.

The growing body of work on bacteria-based therapies[66–69] and the increasing clinical acceptance of FUS[60,61,70,71] provide FUS-actuated bacterial therapeutics a path to ultimate clinical implementation. Potential disease targets include cancers with readily identified primary masses that are challenging to resect surgically, such as head-and-neck, ovarian, pancreatic or brain tumors. FUS-actuated bacterial therapeutics could be also relevant to metastatic tumors since microbial therapy in a single tumor mass can generate a strong adaptive immune response leading to the elimination of distant tumor lesions through the abscopal effect[72]. While EcN cells colonizing A20 tumors have been shown to not disseminate out of tumors into healthy organs[72], other therapeutic models have been described that could foster the dissemination of therapeutic microbes to secondary tumor sites while avoiding healthy organs[73]. Such models would further increase the therapeutic efficacy of locally activated therapeutic microbes beyond the primary tumor. However, further work will be needed to optimize the timing, dose and molecular identity of FUS-activated therapy release for each application. To enhance therapeutic efficacy, it may be beneficial to combine FUS-activated bacterial therapeutics with other molecular or cellular therapies. For example, engineered bacteria and immune cells have distinct and often complementary tumor entry and engraftment profiles. Engineering microbes that successfully enter immunosuppressed tumor regions to secrete checkpoint inhibitors or cytokines could help make this environment more accessible to engineered T cells. In this way, the bacteria and T cells can synergistically exert their therapeutic function from the inside-out and from the outside-in, respectively. Beyond tumor therapy, locally activated bacterial agents have potential utility in a wide array of other biomedical applications. For example, FUS-controlled state switches could be useful in controlling the activity of gut microbes in vivo[74], the function of cell-based living materials in vitro[75–78], and in industrial metabolic engineering[27,79].

## Methods

All animal procedures were performed under a protocol approved by the California Institute of Technology Institutional Animal Care and Use Committee (IACUC).

**Plasmid construction and molecular biology**. All plasmids were designed using SnapGene (GSL Biotech) and assembled via reagents from New England Biolabs for KLD mutagenesis (E0554S) or Gibson Assembly (E2621L). After assembly, constructs were transformed into NEB Turbo (C2984I) and NEB Stable (C3040I) *E. coli* for growth and plasmid preparation. The Bxb1 recombinase-encoding gene was a kind gift of Richard Murray (Caltech). Integrated DNA Technologies synthesized other genes and all PCR primers. Plasmids containing the αCTLA-4, αPD-L1, and Axe-Txe genes were kind gifts of Tal Danino (Columbia University).

**Preparation of cell lines for in vitro and in vivo experiments**. Plasmids containing engineered genetic circuits were transformed into Nissle 1917 *E. coli* (Mutaflor®). Nissle cells were cultured in LB broth (Sigma) and grown on LB agar plates (Sigma) containing appropriate antibiotics. Singular colonies were picked into LB broth and grown overnight in a shaking incubator (30 °C, 250 rpm). The next day, optical density measurements ($OD_{600}$) were taken, and the saturated cultures were diluted to 0.1 $OD_{600}$. Diluted cultures were then allowed to grow to exponential phase until they reached 0.6 $OD_{600}$ before starting assays. Optical density measurements were taken using a Nanodrop 2000c (Thermo Scientific) in cuvette mode.

**Western blot**. Five milliliters of cell media were collected for each sample and concentrated with an Amicon® Ultra-15 Centrifugal Filter Unit. Concentrated cell media was then mixed with Laemmli loading buffer and BME before loading into a pre-cast polyacrylamide SDS-PAGE gel (Bio Rad) and ran at 75 V for 140 min. Western blotting was performed using the Transblot Turbo apparatus and nitrocellulose membrane kit (Bio Rad). Transfer was performed at 25 V for 7 min. Membranes were blocked with 5% Blotto milk (Santa Cruz Biotechnology) in 0.05% TBS-Tween for 1 h at room temperature. Primary staining was performed using a 1:200 dilution of the mouse anti-His sc-8036 antibody (Santa Cruz Biotech) overnight at 4 °C. Blots were then washed three times for 15 min at 4 °C with 0.05% TBS-Tween and stained for 4 h with a 1:1000 dilution of the mouse IgG kappa binding protein (m-IgGκ BP) conjugated to Horseradish Peroxidase (HRP) (Santa Cruz Biotech, sc-516102) at room temperature. After three 15-minute washes, HRP visualization was performed using Super signal west Pico PLUS reagent (Thermo Fisher Scientific). Imaging was performed in a Bio-Rad ChemiDoc MP gel imager. A subsequent epi white light image of the blot under the same magnification was acquired to visualize the stained molecular weight standards.

**Thermal regulation assay**. Once bacterial cell cultures reached approximately 0.6 $OD_{600}$, 50 μL aliquots of each sample were transferred into individual Bio-Rad PCR strips with optically transparent caps and subsequently heated in conditions specific to the experiment using a Bio-Rad C100 Touch thermocycler with the lid set to 50 °C. Following heating, cells continued to incubate overnight undisturbed at either 30 °C (Fig. 1) or 37 °C (Figs. 2–4). The PCR strips were then removed, vortexed, and spun down, and the green fluorescence of each of the samples was measured using the Strategene MX3005p qPCR (Agilent) and an unamplified FAM filter. To measure cell density, the samples were diluted 1:4 with fresh LB media (without antibiotic) and then transferred into individual wells of a 96-well plate (Costar black/clear bottom). Optical density measurements were taken using the SpectraMax M5 plate reader (Molecular Devices). In order to quantify the temperature-dependent gene expression ($E$) using background-subtracted, OD-normalized fluorescence (Figs. 1b–d, f, 2c), Eq. (1) was used:

$$E = \frac{F_{sample}}{OD_{sample}} - \frac{F_{blank}}{OD_{blank}} \tag{1}$$

In this equation, we define $F$ as the raw fluorescence measurement and $OD$ is the $OD_{600}$ measurement of the sample. The value of the blank fluorescence and

blank optical density was determined as the average of $n = 4$ samples of untransformed Nissle cells, as opposed to engineered Nissle cells, in LB. Samples with normalized $F_{sample} < F_{blank}$ were recorded as $E = 0$.

**Screens to optimize circuit behavior**. To improve Bxb1 thermal regulation, a sequence randomized library of the RBS, start codon, and ssrA degradation tag was ordered from Integrated DNA Technologies. PCR products that included the Bxb1 coding region and immediately surrounding sequences were amplified using custom primers and were inserted into the backbone of the rest of the parent plasmid using Gibson Assembly (Fig. 2b–d). This library was transformed into EcN and plated on LB Agar plates with antibiotic resistance at a low colony density of approximately 30 colonies per petri dish. Following overnight incubation at 30 °C to allow the colonies to become visible, these plates were then replicated into two daughter petri dishes using a replica-plating tool (VWR 25395-380). The parent petri dish was incubated at 4 °C until the conclusion of the experiment. One daughter plate was grown overnight at the baseline temperature of 37 °C, and the other was incubated at 42 °C for 1 h and then moved to 37 °C overnight. After colonies became visible, the plates were imaged using a 530/28 nm emission filter to determine colonies that were fluorescent at the 'on' temperature but not at the 'off' temperature (Bio-Rad ChemiDoc MP imager). Promising library variants were then picked from the corresponding parent petri dish at 4 °C and analyzed against the parent plasmid of the library using the liquid culture fluorescence-based assay described above.

**Percent switching assay**. Strips of liquid bacteria samples were prepared and incubated in the Bio-Rad Touch thermocycler. After the prescribed thermal stimulus and incubation at 37 °C, PCR strips were removed, vortexed, and spun down on a tabletop centrifuge. Five 1:10 serial dilutions in liquid LB were then performed, transferring 10 μL of sample into 90 μL of LB media sequentially. After thorough mixing, 50 μL of the most diluted samples was plated onto an LB plate and allowed to incubate at 30 °C overnight. Upon the appearance of visible colonies, plates were imaged using the same Bio-Rad ChemiDoc MP imager with both blue epifluorescence illumination and the 530/28 nm emission filters. The percentage of colonies in the 'on state' ($P$) was determined according to Eq. (2):

$$P = \frac{\text{number of GFP positive colonies counted on a plate}}{\text{total number of colonies counted on a plate}} \quad (2)$$

**Flow cytometry**. EcN cells were incubated for 24 h before assaying with a flow cytometer (MACSQuant VYB) that was thoroughly cleaned to ensure that there are no counts being detected from debris. EcN cells were resuspended in cold PBS + 0.5% BSA (filtered with a 0.2 micron filter) to prevent clumping and were run at 3 different dilutions (targeting 1e6, 1e7, and 1e8 cells/mL).

**Animal procedures**. Female BALB/cJ mice aged 8-12 weeks were purchased from Jackson Laboratory for use in these in vivo experiments. All core facilities are maintained specific-pathogen free. Rodents are housed in HEPA-filtered micro-isolator caging systems at up to five mice per cage. Environmental conditions, such as temperature, humidity and lighting are regulated and monitored for proper level and consistency. A temperature probe is in at least one exhaust air vent per room and is monitored 24 h a day. Water purified by reverse osmosis (RO) is provided ad libitum to rodents via water bottles. Feed (PicoLab Rodent Diet 5053) is received irradiated to eliminate harmful bacteria and viruses and is provided ad libitum via stainless steel wire cage lids with integral feeders. Environmental specifications for mouse housing rooms, including maximum and minimum temperature and humidity values are recorded daily by a member of the animal care staff using a hand-held, battery operated, digital temperature monitoring device. Room temperature is maintained between 71 and 75 degrees F according to the animals' physiological needs, and humidity is maintained between 30-70%. Lighting cycle for all facilities is 13 h on & 11 h off (light cycle 6:00AM - 7:00PM). All air provided to the core animal facilities is filtered and non-recirculated. Animal facilities at Caltech are on-site and are fully accredited by the Association for the Assessment and Accreditation of Laboratory Care International (AAALAC). All experiments involving animals are reviewed by the Caltech Animal Care and Use Committee and are subject to annual review. To establish A20 tumor models in mice, $5 \times 10^6$ A20 cells [TIB-208, ATCC] were collected and suspended in 100 μL phosphate buffered saline (PBS) prior to subcutaneous injection into the flank of each mouse. When tumor volumes reached approximately 100 mm³, engineered EcN cells prepared according to the procedure outlined in the "Preparation of cell lines for in vitro and in vivo experiments" section above were then collected by centrifugation (3000 g for 5 min), washed with PBS three times, and diluted in PBS to 0.625 OD$_{600}$. 100 μL of the resulting solution was injected into each of the A20 tumor bearing mice via tail vein. For some conditions, mice were injected with a combination of αCTLA-4 (200 μg per mouse) and αPD-L1 (100 μg per mouse) checkpoint inhibitors intraperitoneally. These murine checkpoint inhibitors (αCTLA-4 clone 9D9 and αPD-L1 clone 10 F.9G2) were obtained from BioXCell. For thermal actuation using ultrasound, mice were anesthetized using a 2% isoflurane-air mixture and placed on a dedicated animal holder. Anesthesia was maintained over the course of the ultrasound procedure using 1–1.5% isoflurane,

adjusted in real-time to maintain the respiration rate at 20-30 breaths per minute. Body temperature was continuously monitored using a fiber optic rectal thermometer (Neoptix). When appropriate, the target flank was thermally activated using the automated FUS setup described below, cycling between the temperatures of 43 °C and 37 °C every 5 min for 1 h of total heating. Following ultrasound treatment, the mouse was returned to its cage and the size of its tumor was measured with a caliper to track the therapeutic efficacy. When the tumors reached ~1000 mm³ (below the 1500 mm³ maximal tumor size/burden permitted by our animal protocol) mice were culled and the tumors were collected for analysis. Mice that did not have microbial cells in their tumors were excluded from the study.

**Tumor and organ analysis**. Tumors and organs (liver and spleen) were collected and homogenized in ten milliliters of PBS containing 2 mg/ml collagenase and 0.1 mg/ml DNase for one hour at 37 °C. Homogenized tissues were serially diluted and plated onto LB plates with antibiotic selection to quantify the number of cells colonizing the tissues. The percentage of cells activated within tissues was determined by counting the number of GFP positive cells.

**Feedback-controlled focused ultrasound**. We developed a closed loop thermal control setup to maintain a specified predetermined temperature within the tumor of a mouse by modulating the intensity of the FUS. This setup includes a water bath filled with pure distilled water that is being actively cleaned and degassed with an AQUAS-10 water conditioner (ONDA) and maintained at 33 °C with a sous vide immersion cooker (InstantPot Accu Slim). A tumor-bearing mouse that has been anesthetized as described above is fastened nose up vertically to an acrylic arm that is connected to a manual 3D positioning system (Thorlabs) to enable 3D translation of the mouse within the water bath. A Velmex BiSlide motorized positioning system is used to submerge and position the 0.67 MHz FUS transducer (Precision Acoustics PA717) such that the focal point of the transducer lies within the tumor of the mouse. A signal generator (B&K #4054B) generates the thermal ultrasound signal which is then amplified (AR #100A250B) and sent to the ultrasound transducer. The water in this chamber acts as the coupling medium to transfer the ultrasound wave from the transducer to the tumor. To measure the internal tumor temperature during a heating session we temporarily implant a thin fiber optic temperature probe (Neoptix, T1) into the tumor. The custom-ordered probe has a sensing tip with a diameter of 400 μm and length of <2 mm. To insert the probe into tumors we followed the following procedure, as illustrated in Supplementary Fig. 8: (1) shave the tumor before the mouse is placed on a holder, (2) insert a 25-gauge needle into the tumor to guide the fragile thermal probe, (3) insert the fiber-optic probe into the path created by the needle and secure the probe with duct tape. This temperature readout is also used to align the focus of the transducer with the tumor by emitting a constant test thermal ultrasound signal. Once the system is aligned, we run a MATLAB closed loop thermal control script that regulates the signal generator output. Feedback for the controller is provided by the temperature measurements acquired with a sampling rate of 4 Hz. The actuator for the controller is the voltage amplitude of the continuous sinusoidal signal at 0.67 MHz used to drive the FUS transducer, where the voltage is adjusted also at 4 Hz. The transducer acoustic pressure had to be slightly adjusted for each mouse, but typically stayed between 0.6-0.7 MPa (free field). The acoustic pressure field generated by the 670 kHz transducer near its acoustic focus was measured using a Precision Acoustics low frequency-calibrated fiber-optic hydrophone attached to a Velmex X slide translatable stage. The fiber optic hydrophone was in a water bath degassed using an Onda water conditioning system at room temperature. The peak negative pressure was measured at the focal point to be 0.6 MPa given a driving voltage of 35 V peak-to-peak. Full width at half-maximum pressure was measured to be 3.5 mm in the transverse direction and 35 mm in the longitudinal direction (Supplementary Fig. 7, a, b). The system uses a PID controller with anti-windup control that modifies the amplitude of the thermal ultrasound waveform to achieve a desired temperature in the targeted tissues. The Kp, Ki, Kd, and Kt parameters for the PID and anti-windup were tuned using Ziegler-Nichols method, and in some cases adjusted further through trial-and-error tuning to achieve effective thermal control.

**Statistics and replicates**. Data is plotted and reported in the text as the mean ± S.E.M. Sample size is $n = 4$ biological replicates in all in vitro experiments unless otherwise stated. This sample size was chosen based on preliminary experiments indicating that it would be sufficient to detect significant differences in mean values. $P$ values were calculated using a two-tailed unpaired $t$-test.

**Reporting summary**. Further information on research design is available in the Nature Research Reporting Summary linked to this article.

## Data availability

All data are available within the article, supplementary information or the source data file provided with this paper. Source data are provided with this paper.

## Code availability

Custom code used to operate the focused ultrasound system is available on a GitHub repository [https://github.com/drmittelstein/thermal_control].

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

## Acknowledgements

The authors thank Tal Danino, Tiffany Chien, Candice Gurbatri, Sreyan Chowdhury, Dan Piraner and Victoria Hsiao for sharing reagents and helpful discussions. Figures [3a, 3c, 4a, 4b and Supp Fig. 7c] were created with BioRender.com. This research was funded by the Sontag Foundation, the Army Institute for Collaborative Biotechnologies (W911NF-19-D-0001) and the Defense Advanced Research Projects Agency (D14AP00050). M.H.A. and M.T.B. were supported by the NSF graduate research fellowship. M.H.A. was also supported by the Paul and Daisy Soros Fellowship for New Americans. A.B-Z. was supported by the European Union's Horizon 2020 research and innovation programme under the Marie Skłodowska-Curie grant agreement No. 792866. Related research in the Shapiro laboratory is supported by the Burroughs Welcome Career Award at the Scientific Interface, the Packard Foundation Fellowship in Science and Engineering, the Pew Scholarship in the Biomedical Sciences, and the Heritage Medical Research Institute. M.G.S. is an investigator of the Howard Hughes Medical Institute.

## Author contributions

M.H.A. and M.G.S. conceived the study. M.H.A., M.S.Y., D.R.M., M.B.S., A.L-G. and M.T.B. planned and performed experiments. D.R.M. wrote the MATLAB script for in vivo thermal control. A.B-Z. and D.R.M. helped with building the ultrasound heating setup. M.T.B and P.B-L. assisted with performing experiments during the review process. M.H.A. and M.S.Y. analysed data. M.H.A., M.S.Y., D.R.M. and M.G.S. wrote the manuscript with input from all other authors. M.G.S. supervised the research.

## Competing interests

The authors declare no competing interests.
