## [Peer Review File · Nature Communications]

Reviewers' Comments:

Reviewer #1:

Remarks to the Author:

In their manuscript Abedi et al. describe the establishment of a bacterial strain based on *E. coli* Nissle1917 that allows the remote activation of therapeutic genes that have colonized solid murine tumors. They employ a bacterial temperature sensitive transcriptional inhibitor that is active at 37°C and inactive at 42°C. Heat is introduced by targeted focused ultrasound. For long lasting expression they introduce a regulated integrase that is activating a promoter. By additionally manipulating the expression system they find a particular strain that is almost completely silent at 37°C and show an over 100-fold induction at 42°C. Using this expression system the authors express check point inhibitory nanobodies into these bacteria and test them with a murine transplantable tumor. Unfortunately, the results are only suggestive for a future successful bacterial therapy. All together, the underlying idea is highly original and very interesting. However, the manuscript that is obviously directed towards a general audience appears to be very hastily written and needs serious editing. In addition, the *in vivo* experiments were carried out only once and the results could be interpreted as ambiguous. Therefore, more work is required before publication is possible. Points that need to be addressed in detail:

1. Introduction line 16: several chemical inducers have been described before. Such publications should be quoted, like Becker et al, Loessner et al, publications of the Forbes group etc.
2. Introduction line 18: irradiation should be mentioned (Nuyts et al.).
3. Introduction line 28: ts-transcription factors is imprecise because repressors were tested. This is confusing. Please correct.
4. Results: since the manuscript is directed to a general audience to parts of the expression system and their rational need to be shortly introduced.
5. Results line 45: what are these repressors normally inhibiting and where are they derived from. Provide a ref.
6. Results line 48: Explain the rational of the expression system used here.
7. Results line 78: what is the normal function of Bxb1 and where is it derived from. Provide a ref.
8. Results line 81: what is the normal function of the P7 promoter and where is it derived from. Provide a ref.
9. Results line 84: why is the inversion stopped at this state or is the result a 1:1 inversion state.
10. Results line 91: what is the normal function of *ssrA* and where is it derived from. Provide a ref.
11. Results line 107/108: what is the rationale behind increasing the copy number of the plasmid.
12. Results line 109: what is the normal function of the ts-terminator and where is it derived from. Provide a ref.
13. Results line 134: what is the normal function of *pelB* and where is it derived from. Provide a ref. How does the proteins reach the supernatant from the periplasma.
14. Results line 136. Axe-Txe where is it derived from.
15. Results lines 140-150: how is the bacterial physiology affected when the recombinant proteins are expressed. At least growth curves and viability should be shown after induction.
16. Results: only a 1:1 mixture was tested in Fig. 4. Bacteria bearing either one of the constructs should also be tested.
17. Fig. 4d: bacteria from other sites then the tumor should be analyzed for activation. The bacteria might still be disseminating after induction.
18. Fig. 4: these experiments were carried out only 1x. This is not acceptable. Individual growth curves should be shown. At the moment, the data are very much selected and therefore are biased. A mouse that seems to have rejected the tumor should be followed for long term monitoring.

Reviewer #2:

Remarks to the Author:

In view of some limitations of existing tumor immunotherapy, the author proposes a new concept to solve this limitation, and provides feasible methodology and preliminary conceptual evidence. However, the authors did not give sufficient evidence of the safety of this new method compared to traditional treatments. I think the current data and evidence of the manuscript cannot meet the standards published in Nature communication. The author should provide factual evidence that this

method brings sufficient advantages over traditional treatment methods.

Reviewer #3:

Remarks to the Author:

In this work, the authors engineer therapeutic bacteria controlled by focused ultrasound (FUS) for cancer immunotherapy. In the associated circuit, FUS-induced temperature spikes trigger the derepression of TcI42, a temperature-dependent transcriptional repressor. This allows the expression of Bxb1, a serine integrase, which inverts and activates a P7 promoter in front of a therapeutic payload. By this mechanism, transient FUS activation can induce the long-term expression of an anti-tumor therapeutic. After ensuring that the bacteria could express a therapeutic payload, the bacteria were tested in a mouse cancer model.

The system developed in this work represents a new combination of previously-characterized components. The temperature-dependent transcriptional repressors were described in a 2016 paper from the same lab (doi:10.1038/nchembio.2233). This paper extends on this work and applies FUS in conjunction with these repressors to induce GFP expression in *E. coli* that had been subcutaneously injected into a mouse. The novelty of this work is the combination of this prior work with integrase-based switches for the long-term expression of an anti-cancer therapeutic payload.

This paper is appropriate for Nature Communications and I recommend publication after revision.

Major points:

- The novelty of the paper is the testing of the bacteria in a mouse tumor model, but the results are mixed. One of the six FUS-treated tumors disappeared because of treatment. Bacteria in two of the six FUS-treated tumors failed to activate. The remaining three tumors saw less growth than controls. I am not sure how to interpret these data with respect to the viability of this approach. How do comparative treatments perform in this mouse tumor model?
- The ex vivo characterization of the genetic circuits relies on division by OD as a surrogate for cell count, which can lead to artifacts in the interpretation of the data. Either the relationship between OD and viable cell counts should be established (and the latter used for normalization) or these experiments should be verified with flow cytometry or microscopy.
 - o Part of the concern of this normalization is that the induction of the circuits – notably with Temperature – is strikingly small at ~4-fold. It is unclear to what extent this relies on the OD normalization.
 - o The high OD-normalized fluorescence of TcIWT \rightarrow at 41°C in Fig. 1b is unexpected.
- The abstract claims the possibility of spatial control, presumably relative to other treatments, but I do not see evidence in the paper to support this claim.

Minor points:

- There are several asterisks on the plot of Fig. 4c with no accompanying explanations.
- The leftmost section of Fig. 2a (P7 promoter inversion) is unclear and potentially erroneous (the triangles in both should be oriented either outward or inward).
- The rational design (Fig. 2f) was a little bit unclear, specifically how increasing the copy number of the plasmid reduced baseline activation.
- There was an instance of repetitive wording in line 157 (justification of the 43°C focal temperature) since the idea was already expressed earlier in line 143.

Response to Reviewer Comments and Editorial Requests

We thank the reviewers for their overall enthusiasm and helpful comments, which have helped improve the manuscript. Author responses are provided below in blue font. Revised sections of the manuscript are also highlighted in blue. New results are provided in Fig. [4c, 4d] and Supplementary Fig. [1, 3, 4, 5]

Referee #1

In their manuscript Abedi et al. describe the establishment of a bacterial strain based on *E. coli* Nissle1917 that allows the remote activation of therapeutic genes that have colonized solid murine tumors. They employ a bacterial temperature sensitive transcriptional inhibitor that is active at 37°C and inactive at 42°C. Heat is introduced by targeted focused ultrasound. For long lasting expression they introduce a regulated integrase that is activating a promoter. By additionally manipulating the expression system they find a particular strain that is almost completely silent at 37°C and show an over 100-fold induction at 42°C. Using this expression system the authors express check point inhibitory nanobodies into these bacteria and test them with a murine transplantable tumor. Unfortunately, the results are only suggestive for a future successful bacterial therapy. All together, the underlying idea is highly original and very interesting. However, the manuscript that is obviously directed towards a general audience appears to be very hastily written and needs serious editing. In addition, the in vivo experiments were carried out only once and the results could be interpreted as ambiguous. Therefore, more work is required before publication is possible.

Thank you for supporting the originality and general interest of this work and for your helpful suggestions for improvement. We made major revisions to the manuscript in response to these comments, as described under the specific points below.

Points that need to be addressed in detail:

1. Introduction line 16: several chemical inducers have been described before. Such publications should be quoted, like Becker et al, Loessner et al, publications of the Forbes group etc.

Thank you for this suggestion. We have updated the manuscript (Line 16) to reflect these suggested citations and more thoroughly acknowledge the contributions of foundational work in this field.

2. Introduction line 18: irradiation should be mentioned (Nuyts et al.).

Thank you for this suggestion. We have updated the manuscript to discuss this approach (Line 18-22).

3. Introduction line 28: ts-transcription factors is imprecise because repressors were tested. This is confusing. Please correct.

Thank you for catching this issue. We have updated the manuscript to ensure these molecules are referred to as repressors (Line 24&29).

4. Results: since the manuscript is directed to a general audience to parts of the expression system and their rational need to be shortly introduced.

Thank you for this suggestion. We have updated the manuscript Results section to discuss the specific parts of the expression system and their rationale in more detail (Line 84-101). We hope that this will help introduce the readers to all the circuit components.

5. Results line 45: what are these repressors normally inhibiting and where are they derived from. Provide a ref.

Thank you for this suggestion. We have updated the manuscript to explain where these repressors come from and their native function (Line 47-51).

6. Results line 48: Explain the rationale of the expression system used here.

Thank you for this suggestion. We have updated the manuscript to explain why this expression system was used (Line 55-56).

7. Results line 78: what is the normal function of Bxb1 and where is it derived from. Provide a ref.

Thank you for this suggestion. We have updated the manuscript to explain why this expression system was used (Line 85-89).

8. Results line 81: what is the normal function of the P7 promoter and where is it derived from. Provide a ref.

Thank you for this suggestion. We have updated the manuscript to explain where this promoter comes from and why it was chosen (Line 99-101).

9. Results line 84: why is the inversion stopped at this state or is the result a 1:1 inversion state.

Thank you for raising this question. Serine integrases, including Bxb1, modify the recognition sequence while inverting DNA. This new modified sequence is not recognized by Bxb1 and therefore the inversion state is permanent. We have updated the manuscript to ensure that this point is clear to readers (Line 96-98).

10. Results line 91: what is the normal function of *ssrA* and where is it derived from. Provide a ref.

Thank you for this suggestion. We have updated the manuscript to explain where this peptide tag is derived from and added a corresponding reference (Line 107-109).

11. Results line 107/108: what is the rationale behind increasing the copy number of the plasmid.

Thank you for this question. This choice was based on a preliminary experiment where we tested three origins and saw that a medium copy origin would reduce leakage. However, other origins might work as well or better under different conditions and would be good to explore in future experiments. We are not including these experiments in the manuscript because we feel that there is insufficient evidence to make a definitive statement regarding which origin should be used and why.

12. Results line 109: what is the normal function of the ts-terminator and where is it derived from. Provide a ref.

Thank you for this question. We have updated the manuscript to include this information and a corresponding reference (Line 128-131).

13. Results line 134: what is the normal function of pelB and where is it derived from. Provide a ref. How does the proteins reach the supernatant from the periplasma.

Thank you for this question. We have updated the manuscript to include this information and a corresponding reference (Line 158-160). While PelB has been successfully used by us and others to secrete proteins from microbes (cited on Line 160), the underlying mechanism by which proteins exit the periplasm is unclear.

14. Results line 136. Axe-Txe where is it derived from.

Thank you for this suggestion. We have updated the manuscript to include this information and a corresponding reference (Line 162-163).

15. Results lines 140-150: how is the bacterial physiology affected when the recombinant proteins are expressed. At least growth curves and viability should be shown after induction.

Thank you for this question. We performed an additional experiment to evaluate the effect of protein expression from our circuit on bacterial physiology (Supplementary Fig. 3, Line 166-168). Since loss of protein expression is correlated with metabolic burden, we tracked protein expression from our circuit over time and saw no signs of mutational escape.

16. Results: only a 1:1 mixture was tested in Fig. 4. Bacteria bearing either one of the constructs should also be tested.

Thank you for this suggestion. We decided to pursue the 1:1 mixture condition since this condition was previously shown to result in the strongest immune response when compared to administering each therapeutic separately with our tumor model (Ref #10 in the manuscript). Considering that the focus of our paper is on developing a strategy for spatial remote control rather than new therapeutic outputs, we believe this literature-based combination is well-justified and serves our purpose of demonstrating ultrasound-targeted therapeutic functionality.

17. Fig. 4d: bacteria from other sites then the tumor should be analyzed for activation. The bacteria might still be disseminating after induction.

Thank you for this suggestion. We added a new experiment to assess bacterial activation levels in bystander organs (Supplementary Fig. 5, Line 196-199). These results show that we get targeted activation in tumors that is well above the activation levels in the liver and spleen. As for the question on bacterial dissemination, Chowdhury et al. (Ref #71 in the manuscript) have nicely shown that activated EcN do not migrate to other organs or tumors.

18. Fig. 4: these experiments were carried out only 1x. This is not acceptable. Individual growth curves should be shown. At the moment, the data are very much selected and therefore are biased. A mouse that seems to have rejected the tumor should be followed for long term monitoring.

We apologize for the confusion. Our original data was actually the result of three separate experiments (involving multiple animals) conducted with fresh cells each time. To further reinforce our findings, we conducted the experiment a fourth time and supplemented the existing data with these results (Fig. 4c, 4d). At least eight mice were analyzed for each control condition and ten mice were analyzed for therapeutic condition. This has been clarified in the main text (Fig. 4 legend). Results from individual animals have been added in Supplementary Fig.4 as requested.

Referee #2

In view of some limitations of existing tumor immunotherapy, the author proposes a new concept to solve this limitation, and provides feasible methodology and preliminary conceptual evidence. However, the authors did not give sufficient evidence of the safety of this new method compared to traditional treatments. I think the current data and evidence of the manuscript cannot meet the standards published in Nature communication. The author should provide factual evidence that this method brings sufficient advantages over traditional treatment methods.

Thank you for your review and constructive feedback. Based on your critique and comments from the other reviewers, we have updated the manuscript to provide additional *in vivo* experimental data demonstrating the efficacy, repeatability and safety of our approach. In particular, we added new data in Supplementary Fig. 5 showing that our therapeutic microbes are being activated specifically in ultrasound-targeted tumors, with minimal activation within bystander organs. This addresses the major potential safety concern related to off-target activity of the established approach to microbial tumor therapy. Previous studies have already demonstrated the overall rationale and advantages of tumor-homing bacterial approaches to cancer treatment, as clarified in lines 7-9 of the revised manuscript.

In addition, we bolstered the efficacy results in Fig. [4c,4d] with additional data and characterized the impact of our protein expression circuits on the probiotic agent (Supplementary Fig. 3).

With these additions, we are certain that this manuscript represents a major conceptual advance backed by solid data that will be of interest to a broad audience.

Referee #3

In this work, the authors engineer therapeutic bacteria controlled by focused ultrasound (FUS) for cancer immunotherapy. In the associated circuit, FUS-induced temperature spikes trigger the derepression of Tc142, a temperature-dependent transcriptional repressor. This allows the expression of Bxb1, a serine integrase, which inverts and activates a P7 promoter in front of a therapeutic payload. By this mechanism, transient FUS activation can induce the long-term expression of an anti-tumor therapeutic. After ensuring that the bacteria could express a therapeutic payload, the bacteria were tested in a mouse cancer model.

The system developed in this work represents a new combination of previously-characterized components. The temperature-dependent transcriptional repressors were described in a 2016 paper from the same lab (doi:10.1038/nchembio.2233). This paper extends on this work and applies FUS in conjunction with these repressors to induce GFP expression in *E. coli* that had been subcutaneously injected into a mouse. The novelty of this work is the combination of this prior work with integrase-based switches for the long-term expression of an anti-cancer therapeutic payload.

This paper is appropriate for Nature Communications and I recommend publication after revision.

Thank you for your positive review and helpful comments. We have substantially revised the manuscript in response to your feedback as under the specific points below.

Major points:

1. The novelty of the paper is the testing of the bacteria in a mouse tumor model, but the results are mixed. One of the six FUS-treated tumors disappeared because of treatment. Bacteria in two of the six FUS-treated tumors failed to activate. The remaining three tumors saw less growth than controls. I am not sure how to interpret these data with respect to the viability of this approach. How do comparative treatments perform in this mouse tumor model?

Thank you for this valuable feedback. To reinforce our findings as suggested by the reviewer, we performed further *in vivo* experiments in which we collected data from an additional cohort of mice in which therapeutic microbes are activated in A20 tumors with focused ultrasound (FUS) (Fig. 4c, 4d). With this additional data, our figure represents the combination of four different experimental runs performed on different days with different mice and different microbes. At least eight mice were analyzed for each control condition and ten mice were analyzed for therapeutic condition. These results clearly demonstrate the ability to specifically activate the microbial therapeutics inside tumors and the marked reduction of tumor growth in mice receiving the combination treatment of therapeutic bacteria + FUS compared to controls (two-way ANOVA analysis where the therapy was compared to each of the controls). These results give us confidence in the robustness of our overall findings.

As the reviewer pointed out, our success rate in FUS-based activation of the cells is not 100%. Indeed, we recorded one additional failure to activate in the new batch of 4 mice. We believe this is a consequence of our FUS setup's inability to fully heat the tumor, meaning that the heated part of the tumor may not match where bacterial colonization has taken hold. Fortunately, this is only a problem for the type of home-built FUS system we have available in our lab. Clinical FUS systems use MRI feedback and dynamic mechanical or electrical focusing to ensure the precise heating of defined volumes, which can be as large as a whole tumor. We have added a discussion of this point, with references to the clinical MRI guided FUS, on lines 203-206.

Regarding tumor disappearance, we only observed this in one animal, and have edited the manuscript to explicitly state that this is an anomaly and not a general representation of our results (Line 201). The main therapeutic outcome is the reduction in tumor growth, which matches the results seen in other successful tumor-homing bacteria studies (e.g. Ref. #10). The primary novelty of our approach is that it adds a mechanism for precisely localizing where microbes are activated. To emphasize this point, we added a new experiment to assess bacterial

activation levels in bystander organs (Supplementary Fig. 5, Line 196-199). These results show that we get targeted activation in tumors that is well above the activation levels in the liver and spleen.

Finally, to help readers better understand our results and the limitations of our approach, we plotted the growth curves of each individual tumor and included the data as a supplementary Fig. 4.

2. The ex vivo characterization of the genetic circuits relies on division by OD as a surrogate for cell count, which can lead to artifacts in the interpretation of the data. Either the relationship between OD and viable cell counts should be established (and the latter used for normalization) or these experiments should be verified with flow cytometry or microscopy.
 - a. Part of the concern of this normalization is that the induction of the circuits – notably with Temperature – is strikingly small at ~4-fold. It is unclear to what extent this relies on the OD normalization.
 - b. The high OD-normalized fluorescence of TcI_{WT} at 41°C in Fig. 1b is unexpected.

Thank you for these comments. To assess the validity of using OD as a surrogate for total cell count, we compared fold-change measurements of TcI42 EcN cells normalized to OD with fold-change measurements normalized to cell counts obtained by flow cytometry (Supplementary Fig. 1a). In these measurements we saw a strong correspondence between results that use OD or cell count for normalization. We confirmed that neither of these quantities is strongly affected by incubation at different temperatures (Supplementary Fig. 1b).

Regarding the small fold change for TcI_{WT}, this is expected when cells are heated for only 60 minutes (as seen in Ref. #37). Additionally, EcN cells tend to produce a higher baseline level of expression at 37 °C when compared to strains used in that previous study, reducing the overall fold change. This modest fold-change for TcI_{WT} motivated our use of the engineered bioswitch TcI42 and the other circuit optimizations performed in this study.

Regarding the 41°C peak in TcI_{WT} expression seen in Fig. 1b, this is consistent with previously published results (Ref. #37). The precise peak depends on the identity of the repressor and the cell strain being used and is not an artifact of normalization.

3. The abstract claims the possibility of spatial control, presumably relative to other treatments, but I do not see evidence in the paper to support this claim.

Thank you for this suggestion. To demonstrate spatial specificity, we added new experimental data in Supplementary Fig. 5 showing that probiotic agents are activated in ultrasound-targeted tumors at much higher rates than in bystander organs.

Minor points:

1. There are several asterisks on the plot of Fig. 4c with no accompanying explanations.

Thank you for catching this omission. We have updated the figure legend to correct it.

2. The leftmost section of Fig. 2a (P7 promoter inversion) is unclear and potentially erroneous (the triangles in both should be oriented either outward or inward).

Thank you for catching this error. We have updated the manuscript to fix it.

3. The rational design (Fig. 2f) was a little bit unclear, specifically how increasing the copy number of the plasmid reduced baseline activation.

Thank you for this question. This choice was based on a preliminary experiment where we tested three origins and saw that a medium copy origin would reduce leakage. We are not including these results in the manuscript because there wasn't enough evidence to make a definitive statement regarding which origin is optimal.

4. There was an instance of repetitive wording in line 157 (justification of the 43°C focal temperature) since the idea was already expressed earlier in line 143.

Thank you for catching this typo. We have removed the redundant wording from the manuscript.

Reviewers' Comments:

Reviewer #1:

Remarks to the Author:

The authors have appropriately handled my points of critique. The manuscript has significantly improved. The authors did an extremely good job by adding the information on strategy and genetic elements used without interfering with the flow of the description. Two very minor points are still open to my mind:

1. The authors should mention in M&M that they have used *E. coli* Nissle 1917 as the carrier strain and possibly also the source. ECN is only mentioned once in the introduction.
2. The authors now also present data on the dissemination of the bacteria from the tumor to the healthy organs or better the lack thereof. They do not test dissemination to a second tumor after intratumoral application and activation of the bacteria in a primary tumor. They argue that the lack of dissemination by ECN was described by other authors already. I find this argument slightly unsatisfactory. I realize that this is not really the scope of the work. On the other hand, the absence of bacteria in the healthy organs could be due to the quick elimination of such microorganisms by the immune system. The immunosuppressive microenvironment in the tumor might lead to a different outcome. Dissemination to a secondary tumor was shown by Kocijancic et al., 2017, although a different bacterial species was used and different transplantable tumors. Here the argument was that under such conditions the bacteria could carry therapeutic substances to secondary tumors or metastases. The system described in the present work would be extremely useful in this respect. The authors might want to discuss this aspect.

Reviewer #3:

Remarks to the Author:

Looks good to me

Reviewer #4:

Remarks to the Author:

This study describes the development of temperature-responsive engineered microbes for tumor-selective expressions of immunotherapeutics. In vitro and in vivo data presented suggested that adding heat of $\sim 42^{\circ}\text{C}$ or greater post microbial therapy achieved enhanced protein expressions and release of checkpoint proteins in the tumor. Overall, the project is innovative, and the design of a thermally activated plasmid is elegant. Key weaknesses of this study are mainly in the requirement of long-duration FUS heating to achieve clinically relevant robust bacterial protein expressions. This may be somewhat challenging to achieve in very large solid tumors even with MR-heating. Comments are given below:

1. For FUS heating, a fiber-optic temperature sensor was utilized for thermometry. Method section can provide more details on the sensor type, and the insertion technique, as the insertion of a probe in a small tumor, can cause tumor bleeding, and improper temperature measurements in a water bath. How was the sensor secured in the somewhat smaller tumors inside the water bath?
2. The authors correctly noted that MR-based methods can be more accurate for thermometry, and the feasibility demonstration using their home-built system may have had some limitations. It may be nice to describe the number of mice that were utilized to generate the temperature map in Fig. 4a. Additionally, the FUS parameters used including the acoustic power, and details on the size of focal spots could have given better insights on FUS tumor coverages.
3. The authors noted that 1/3 of the FUS-treated tumors did not activate the circuit. These non-activated mice were removed from the tumor growth data. This may be alright if the heating was not aligned with the bacterial locations, however, the rationales appear more hypothetical. The scientific community will benefit from some insights on the bacterial localization rates and their probable locations in tumor over 2-days post-IV injections, and how that information may be

utilized to plan FUS treatments of the tumor in a site-specific manner.

Response to Reviewer Comments and Editorial Requests

We thank the reviewers for their overall enthusiasm and helpful comments, which have helped improve the manuscript. Author responses are provided below in blue font. Revised sections of the manuscript are also highlighted in blue. New results are provided in **Supplementary Figures 4,7, and 8**.

Referee #1

The authors have appropriately handled my points of critique. The manuscript has significantly improved. The authors did an extremely good job by adding the information on strategy and genetic elements used without interfering with the flow of the description.

We thank the reviewer for their thoughtful comments and suggestions, which helped improve our manuscript.

Two very minor points are still open to my mind:

1. The authors should mention in M&M that they have used *E. coli* Nissle 1917 as the carrier strain and possibly also the source. ECN is only mentioned once in the introduction.

To make the early mention of *E. coli* Nissle 1917 more prominent, we ensured that our use of it is specified in the introduction (line 29), along with relevant literature references (moved up from later in the manuscript).

2. The authors now also present data on the dissemination of the bacteria from the tumor to the healthy organs or better the lack thereof. They do not test dissemination to a second tumor after intratumorally application and activation of the bacteria in a primary tumor. They argue that the lack of dissemination by ECN was described by other authors already. I find this argument slightly unsatisfactorily. I realize that this is not really the scope of the work. On the other hand, the absence of bacteria in the healthy organs could be due to the quick elimination of such microorganisms by the immune system. The immunosuppressive microenvironment in the tumor might lead to a different outcome. Dissemination to a secondary tumor was shown by Kocijancic et al., 2017, although a different bacterial species was used and different transplantable tumors. Here the argument was that under such conditions the bacteria could carry therapeutic substances to secondary tumors or metastases. The system described in the present work would be extremely useful in this respect. The authors might want to discuss this aspect.

As recommended by the reviewer, we added a discussion of this future research question in the manuscript on line 236-239.

Referee #4

This study describes the development of temperature-responsive engineered microbes for tumor-selective expressions of immunotherapeutics. In vitro and in vivo data presented suggested that adding heat of ~42C or greater post microbial therapy achieved enhanced protein expressions and release of checkpoint proteins in the tumor. Overall, the project is innovative, and the design of a thermally activated plasmid is elegant. Key weaknesses of this study are mainly in the requirement of long-duration FUS heating to achieve clinically relevant robust

bacterial protein expressions. This may be somewhat challenging to achieve in very large solid tumors even with MR-heating. Comments are given below:

Thank you for your positive review and helpful suggestions.

1. For fus heating, a fiber-optic temperature sensor was utilized for thermometry. Method section can provide more details on the sensor type, and the insertion technique, as the insertion of a probe in a small tumor, can cause tumor bleeding, and improper temperature measurements in a water bath. How was the sensor secured in the somewhat smaller tumors inside the water bath?

Thank you for this suggestion. We updated the Methods section to include more information about the probe used and how it was inserted and secured (line 379-383). We also added a new figure (**Supplementary Fig. 8**) with images showing the insertion procedure. During this procedure, we very rarely saw signs of bleeding. In the very rare instance where bleeding was observed, the fiber-optic cable was moved to a different location in the tumor.

Supplementary Figure 8 | fiber-optic probe insertion procedure. To insert the probe into tumors we followed the following procedure: (1) shave the tumor before the mouse is placed on a holder, (2) insert a 25-gauge needle into the tumor to guide the fragile thermal probe, (3) insert the fiber-optic probe into the path created by the needle and secure the probe with duct tape.

2. The authors correctly noted that MR-based methods can be more accurate for thermometry, and the feasibility demonstration using their home-built system may have had some limitations. It may be nice to describe the number of mice that were utilized to generate the temperature map in Fig. 4a. Additionally, the FUS parameters used including the acoustic power, and details on the size of focal spots could have given better insights on FUS tumor coverages.

Thank you for this comment. To demonstrate the reproducibility of our feedback-controlled heating approach, we have added a new figure (**Supplementary Fig. 4**) with heating traces from four additional mice. (Since the heating is applied under feedback control in individual subjects, we believe showing multiple individual traces is more appropriate than averaging across animals.)

Supplementary Figure 4 | FUS oscillatory heating of tumors. Tumor temperature measurements in four individual mice treated with alternating 5-min steps between 37 °C and 43 °C under feedback control.

In addition, we have included further detail on the acoustic parameters in the Methods (line 389-396) and a new figure (**Supplementary Fig. 7**) showing the experimentally measured focal zone of our transducer and comparing it to the size of our tumors.

Supplementary Figure 7 | Characterization of the FUS beam. (a) Normalized peak-negative pressure field of the 670 kHz transducer used in this study, measured using a fiber-optic hydrophone in the transverse plane, orthogonal to direction of propagation. (b) Normalized peak-negative pressure measured in the longitudinal plane, along the direction of FUS propagation. (c) Illustration of the typical cross-sectional dimensions of the tumors treated in this study relative to the area covered by the FUS focal zone (3.5 mm lateral and 35 mm longitudinal full width at half maximum pressure). The illustration includes a hypothetical sub-tumoral distribution of bacterial cells, showing how the ultrasound heating setup used in this study could correctly (left) or incorrectly (right) target the cells for activation.

3. The authors noted that 1/3 of the FUS-treated tumors did not activate the circuit. These non-activated mice were removed from the tumor growth data. This may be alright if the heating was not aligned with the bacterial locations, however, the rationales appear more hypothetical. The scientific community will benefit from some insights on the bacterial localization rates and

their probable locations in tumor over 2-days post-IV injections, and how that information may be utilized to plan FUS treatments of the tumor in a site-specific manner.

Thank you for this comment. We have included a citation from one of our previous publications documenting the spatial distribution of bacteria inside tumors (as shown in the image below).

Image from ref. 64, showing a subcutaneous A20 tumor (outlined) imaged with ultrasound to reveal its vasculature (red, superlocalization imaging) and the location of EcN cells (green, nonlinear imaging of EcN cells expressing acoustic reporter genes). This serves as an example of how bacterial agents colonize the tumor inhomogeneously, tending to localize in the poorly vascularized tumor core.

We added discussion about the overlap between the ultrasound focal zone and the typical bacterial distribution in the tumors (line 209-211), referencing **Supplementary Fig. 7** (shown above). As shown in panel (c) of this new figure, it is possible for the beam to miss the bacteria using our available setup. In actual clinical implementations of this technology, image-guided FUS would comprehensively cover the relevant tumor volume, as commented in the Discussion.

Finally, we have added further analysis to the main text establishing that, even if the non-activated mice remain in the analysis, our treatment achieves a highly statistically significant therapeutic endpoint in tumor growth (line 213-216).

Thank you for your helpful review! We appreciate your time and consideration.

Reviewers' Comments:

Reviewer #4:

Remarks to the Author:

My feedbacks were appropriately addressed by the authors in the revised submission.